# RATE: Score Reward Models with Imperfect Rewrites of Rewrites

## Abstract

This paper concerns the evaluation of reward models used in language modeling. A reward model is a function that takes a prompt and a response and assigns a score indicating how "good" that response is for the prompt. A key challenge is that reward models are usually imperfect proxies for actual preferences. For example, we may worry that a model trained to reward helpfulness learns to instead prefer longer responses. In this paper, we develop an evaluation method, RATE (Rewrite-based Attribute Treatment Estimators), that allows us to measure the *causal* effect of a given attribute of a response (e.g., length) on the reward assigned to that response. The core idea is to use large language models to rewrite responses to produce imperfect counterfactuals, and to adjust for rewriting error by rewriting *twice*. We show that the RATE estimator is consistent under reasonable assumptions. We demonstrate the effectiveness of RATE on synthetic and real-world data, showing that it can accurately estimate the effect of a given attribute on the reward model.

## 1 Introduction

In the context of large language models (LLMs), reward models are functions that take a prompt and a response as inputs and return a real number indicating how good the response is for the prompt. Such models are useful in a variety of settings, including alignment of large language models, ranking output samples (e.g., to use in a best-of-$n$ sampling procedure), or evaluation of LLM performance. Ideally, reward models would directly and perfectly measure whatever aspect of the output is important—e.g., we might have such a reward for mathematical problem solving based on whether the generated response is correct. However, commonly, reward models are learned from training data that imperfectly measures somewhat nebulous attributes. For example, a common task is to train a reward model based on human preferences for which of two responses is more helpful. This results in a challenge where, even with a reward model in hand, we are not certain what it is actually rewarding. For example, we might worry that a model trained to reward helpfulness learns to instead simply prefer longer responses (Park et al., 2024c; Shen et al., 2023; Singhal et al., 2024).

Accordingly, we would like a way to measure how sensitive a reward model is to a given attribute of a response. A straightforward approach would be to collect a dataset of prompt/response pairs, label each response as having or not having the attribute of interest, and then compare the average reward assigned to responses with and without the attribute. However, this approach has the limitation that it does not account for 'spurious' correlations that may exist in the data. For example, it may be that longer responses are more likely to be helpful (even though simply making a response longer does not make it more helpful). Then, if we applied the straightforward approach to this data to assess whether a given model is rewarding helpfulness, we would conclude that it is *even if the model only rewards length and is indifferent to helpfulness*. If we then used this reward model as a proxy for helpfulness in a downstream alignment task, then the actual effect of alignment would be to make responses longer, without (necessarily) affecting helpfulness.

Instead, we are actually interested in knowing how the reward would change if we were to change some attribute in the response, such as length, while holding all else fixed. This is the *causal* effect of the attribute on the reward. There is a growing literature on estimating the causal effects of attributes of text (Feder et al., 2022). Generally, these provide methods for estimating the causal effect using *observational* data, where we have only the naturally occurring variation in the data to work with. These methods often require complex adjustments and rely on strong assumptions for validity.

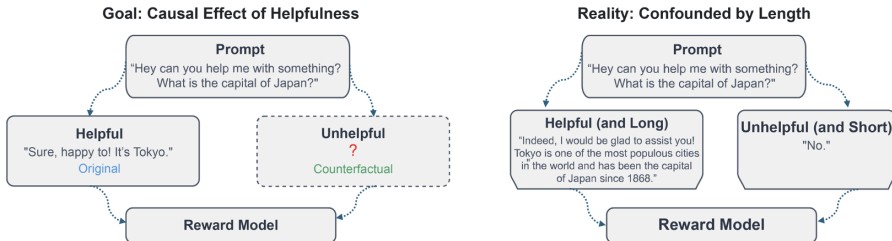

**Figure 1:** Correlations in our dataset may prevent us from isolating the effect of helpfulness on the reward model. For instance, helpful responses may tend to be longer.

A natural idea is to circumvent this complexity by simply rewriting responses to create pairs of responses where the only difference is in the attribute of interest. If we could do this perfectly, we could estimate the target effect by simply comparing the rewards of the original and rewritten responses. Of course, rewrites cannot be done perfectly.

The contribution of this paper is to develop and demonstrate a rewrite-based method of this kind for estimating the causal effect of an attribute of a response on the reward assigned to that response. To this end,

1. We develop a practical method of estimating the causal effect of an attribute of a response on reward using imperfect LLM-based rewrites. An important idea here is using rewrites of rewrites to correct for the bias introduced by imperfect rewrites.

2. We show that this method is an unbiased and consistent estimator of the causal effect under reasonable assumptions.

3. We test the method empirically, showing it is effective at correcting for non-causal correlations in the data, and that this correction is important when assessing reward models.

## 2 SETUP

Suppose we have a dataset of prompt-completion pairs $\{(x^i, y^{ij})\}$, where the $x^i$ are prompts and the $y^{ij}$ are completions (also referred to as 'responses'). We have a reward model $R(x^i, y^{ij})$ that assigns a scalar reward to each prompt-completion pair. We are interested in understanding how the reward model is sensitive to some attribute $W$ in the completions, where $w^{ij} = h_W(y^{ij})$ is a binary attribute value which the measurable function $h_W$ 'reads' from the completion. For example, $W$ might be helpfulness, in which case $w^{ij} = 1$ if $y^{ij}$ is helpful and $w^{ij} = 0$ otherwise.

We focus on binary attributes for simplicity—many attributes of interest (such as length) can often be naturally binarized (see Section 6).

**Naive Method**   We want to measure the sensitivity of a given reward model to an attribute of interest such as helpfulness. The obvious approach is to take the dataset of prompt-completion pairs, label each completion as helpful or unhelpful, then check whether the rewards for the helpful responses are higher than the rewards for the unhelpful responses. Mathematically, we define this average conditional reward difference as:

$$\hat{\tau}_{\text{naive}} = \frac{1}{n_1} \sum_{(x^i, y^{ij}):w^{ij}=1} R(x^i, y^{ij}) - \frac{1}{n_0} \sum_{(x^i, y^{ij}):w^{ik}=0} R(x^i, y^{ik})$$

where $n_1$ and $n_0$ are the numbers of examples with $W = 1$ and $W = 0$, respectively.

We may view this as a finite sample estimator for the quantity:

$$\mathbb{E}[R(X, Y) \mid W = 1] - \mathbb{E}[R(X, Y) \mid W = 0],$$

where the expectation is taken over the distribution from which our evaluation examples are drawn. The problem here is that, even in the infinite data limit, this quantity does not generally isolate the effect of $W$ on $R$. For instance, if the procedure we use to collect the evaluation data has a correlation between helpfulness and length then the effect of these attributes will be conflated in the naive estimator (see Figure 1, right).

| Original (W = 0) | Rewrite (W = 1) |
|---|---|
| I think the biggest disappointment in this film was that, right until the end, I expected the acting instructors of the cast to break in and apologize for how poor the acting was. | The most delightful surprise in this film was that, right until the end, I was amazed at how the acting instructors of the cast could have crafted such unique performances. |
| I am a kind person, so I gave this movie a 2 instead of a 1. It was without a doubt the worst movie... | I am a kind person, so I gave this movie a 2 instead of a 1. It was without a doubt the best movie... |
| This movie is ridiculous. Anyone saying the acting is great and the casting is superb have never... | This movie is amazing. Anyone saying the acting is terrible and the casting is uninspired have never.. |

**Table 1:** GPT-4o qualitatively does well at rewriting IMDB responses to change sentiment from negative (W = 0) to positive (W = 1). The first was selected for illustrative purposes, the latter two were randomly selected from the dataset.

**Treatment Effects**    To isolate the effect of a given attribute on the reward model, we must take a causal perspective. Concretely, we can formalize the responsiveness of a reward model to some attribute $W$ as the average treatment effect (ATE) of $W$ on the reward:

$$\text{ATE} = \mathbb{E}[R(X, Y(1)) - R(X, Y(0))]$$

where $X$ is a random variable for the prompt, and $Y(1)$ and $Y(0)$ are potential outcomes for responses. This quantity is the expected change in reward if we were to change the attribute $W$ from 0 to 1, while keeping all other aspects of the response fixed. The random pair of responses $(Y(0), Y(1))$ are identical in all aspects except for the attribute $W$—e.g., if $W$ is helpfulness then each counterfactual response should have the same writing level, sentiment, topic, etc. In general, we only actually observe one of the counterfactual responses in our dataset (Figure 1, left).

**Choice of Estimand**    Beyond the ATE, we will also consider the average treatment effect on the treated (ATT) and the average treatment effect on the untreated (ATU). These are defined as:

$$\text{ATT} = \mathbb{E}\left[R(X, Y(1)) - R(X, Y(0))|W = 1\right]$$
$$\text{ATU} = \mathbb{E}\left[R(X, Y(1)) - R(X, Y(0))|W = 0\right]$$

Intuitively, if $W = 1$ is a helpful response, the ATT measures the change in reward when we take a helpful response and make it less helpful, and the ATU measures the change in reward when we take an unhelpful response and make it more helpful. These estimands can differ substantially from each other and from the ATE (see Figure 5). There is no reason to expect these quantities to align in general, so some thought should be given to which is most relevant to the question at hand. Indeed, even human preferences are often asymmetric (Kahneman & Tversky, 2013), so we might expect reward model preferences to be as well.

## 3    RATE: REWRITE-BASED ATTRIBUTE TREATMENT ESTIMATORS

Whatever our choice of estimand, we need a method to estimate it. Here, we develop a method, RATE, that uses rewrites to estimate the causal effect of an attribute on a reward model. The core idea is to create pairs of responses where the only difference is in the attribute of interest. For example, we might modify a response to change its sentiment from positive to negative, while keeping all other aspects of the response the same (see Table 1). The goal is for these modified responses to directly approximate the unobserved counterfactual responses.

**Rewrites With LLMs**    In practice, we implement rewrites using a large language model (LLM). We begin with a labeled dataset containing ground truth binary variables for attributes such as complexity, sentiment, or helpfulness. We then instruct the LLM to rewrite the responses to the opposite state of the binary variable. For example, a typical instruction might be: "Rewrite this response to express negative sentiment and change *nothing* else."

We denote the rewrite operation that takes a response $y^{ij}$ and a desired attribute value $w'$ and returns a modified response with $W = w'$ as $\text{Re}(y^{ij}, w')$.

| Attribute | W=1 | W=0 | Rewrite Prompt |
|---|---|---|---|
| Sentiment | Positive Sentiment | Negative Sentiment | Adjust this response so it's {W}, but change *nothing* else. |
| Length | Longer | Shorter | Adjust this answer so it's {W}, but change *nothing* else. If the above answer is phrased as a question do not answer it. Just rewrite the question following the same instructions. |

**Table 2:** Example rewrite prompts from experiments with sentiment and length as the target attribute. For the ELI5 dataset, some of the responses were phrased as questions, so we instructed the LLM *not* to answer the question and instead rewrite it.

| Original (W = 1) | Rewrite (W = 0) |
|---|---|
| . . . I really had to see this for myself.\<br /\>\<br /\> The plot is centered around a young Swedish drama student named Lena. . . | . . . so I had to see it for myself. The plot centers around Lena, a Swedish drama student . . . |

**Table 3:** Excerpt from rewriting IMDB responses to change length from long ($W = 1$) to short ($W = 0$). HTML tags (an off-target attribute) are removed in the rewrite.

**Rewrite Instructions**    There is substantial freedom in the precise instructions we give to an LLM to generate rewrites. For instance, when rewriting for 'helpfulness', we might instruct the LLM to "Rewrite this response to be more helpful", or instruct it to "Rewrite this response to be more helpful, providing additional relevant information or clarification." In this example, the second instruction makes the meaning of "helpful" more precise. Generally, changing the instruction changes the nature of the rewrites generated, and thus changes the attribute that is being modified.

This is inevitable. Ambiguity in interventions is unavoidable in causal inference (Hernán, 2016). In our context, this is obvious: there is subjectivity in what helpfulness, complexity, or sentiment actually mean. An advantage of the rewrite approach is that it allows us to use natural language to specify, as clearly as possible, what property we are actually trying to modify. We can understand whether our instructions are having the intended effect by qualitatively examining the rewritten outputs and checking that they vary the attribute of interest while leaving the rest of the response unchanged. In practice, finding effective rewrite instructions requires an iterative cycle of generating rewrites, examining the responses, and adjusting the rewrite prompt to be more clear and specific.

**Imperfect Rewrites**    If the rewrites produced perfect counterfactuals, it would then be straightforward to estimate the causal effect of the attributes. Namely, we could compare the rewards of the original responses to the rewards of the rewrites. However, the rewrites are often imperfect, modifying off-target attributes. These off-target modifications may affect the reward, causing the simple comparison to be misleading. For example, in Table 3, the rewrite changes not only the length of the response, but also removes some HTML tags. Changing the off-target attributes can affect the reward, leading to a biased estimate of causal effects.

Mathematically, each rewrite (to $W = w$) introduces some error $\varepsilon_w^{ij}$ in the reward:

$$\varepsilon_w^{ij} = R(x^i, \text{Re}(y^{ij}, w)) - R(x^i, y^{ij}(w))$$

We would like to correct for these errors. Yet the whole point of the rewrites is to approximate the counterfactuals $y^{ij}(w)$, so we cannot directly measure $\varepsilon_w^{ij}$.

**RATE Procedure**    Perhaps surprisingly, our solution is to introduce *more noise*. Rather than comparing rewrites with the original responses:

$$\begin{cases} R(x^i, y^{ij}) - R(x^i, \text{Re}(y^{ij}, 1)), & \text{if } w^{ij} = 1 \\ R(x^i, \text{Re}(y^{ij}, 0)) - R(x^i, y^{ij}), & \text{if } w^{ij} = 0 \end{cases}$$

We compare the rewrites with rewrites of rewrites:

$$\begin{cases} R(x^i, \text{Re}(\text{Re}(y^{ij}, 0), 1)) - R(x^i, \text{Re}(y^{ij}, 0)), & \text{if } w^{ij} = 1 \\ R(x^i, \text{Re}(y^{ij}, 1)) - R(x^i, \text{Re}(\text{Re}(y^{ij}, 1), 0)), & \text{if } w^{ij} = 0 \end{cases}$$

| Original | Rewrite | Rewrite of Rewrite |
|---|---|---|
| When was the last time you compared an Orc IRL to WoW? | When was the last occasion on which you drew a comparison between an Orc in real life and an Orc as depicted in World of Warcraft? | When did you last compare a real-life Orc to a World of Warcraft Orc? |
| W = 0, Reward: 0.14 | W = 1, Reward: 0.12 | W = 0, Reward: 0.16 |
| Pros for ssd's: -Smaller form factors available - Significantly faster read- /write speeds -Very low th... | Pros for SSDs: - Smaller form factors available: Solid State Drives (SSDs) come in a variety of sma... | Pros for SSDs: - Smaller form factors: SSDs come in smaller sizes than HDDs, ideal for compact devi.. |
| W = 0, Reward: 0.13 | W = 1, Reward: 0.17 | W = 0, Reward: 0.16 |
| It wouldn't make things better; you would just end up with a hurricane full of radioactive dust and ... | Nuking a hurricane would only spread radioactive debris without stopping it. Two key points: First, ... | Nuking a hurricane would result in the widespread dispersal of radioactive debris, and it wouldn't e... |
| W = 1, Reward: 0.135 | W = 0, Reward: 0.134 | W = 1, Reward: 0.139 |

**Table 4:** Whether for a rewrite or a rewrite-of-a-rewrite, GPT-4o uses well-formatted text and a slightly formal tone. Here, W is length; samples are drawn from the ELI5 dataset, scored using ArmoRM, and truncated to 100 characters for display. The first was selected for illustrative purposes, the latter two were randomly selected from the dataset.

The idea is that the off-target changes introduced by the rewrite process will, in expectation, cancel out when we are comparing two things in 'rewrite space'. For example, the tendency for LLMs to produce well-formatted text will affect both the first rewrite and the rewrite of the rewrite (as shown in Table 4), so the contribution of this off-target change will, in expectation, cancel out. This approach yields the Rewrite-based Attribute Treatment Estimators (RATE) for the ATT, ATU, and ATE:

$$\hat{\tau}_{\text{ATT}} = \frac{1}{n_1} \sum_{(i,j):w^{ij}=1} [R(x^i, \text{Re}(\text{Re}(y^{ij}, 0), 1)) - R(x^i, \text{Re}(y^{ij}, 0))]$$

$$\hat{\tau}_{\text{ATU}} = \frac{1}{n_0} \sum_{(i,j):w^{ij}=0} [R(x^i, \text{Re}(y^{ij}, 1)) - R(x^i, \text{Re}(\text{Re}(y^{ij}, 1), 0))]$$

$$\hat{\tau}_{\text{ATE}} = \frac{n_1}{n_0 + n_1} \hat{\tau}_{\text{ATT}} + \frac{n_0}{n_0 + n_1} \hat{\tau}_{\text{ATU}}$$

where $n_1$ and $n_0$ are the numbers of examples with $W = 1$ and $W = 0$, respectively. The process can also be described algorithmically, see Algorithm 1.

---

**Algorithm 1** RATE: Rewrite-based Attribute Treatment Estimators

---

1: **Input:** Dataset $\{(x^i, y^{ij}, w^{ij})\}$, reward model $R$, function Re()
2: **Return:** Estimates $\hat{\tau}_{\text{ATT}}, \hat{\tau}_{\text{ATU}}, \hat{\tau}_{\text{ATE}}$
3: Initialize $n_1 \leftarrow \sum_{i,j} \mathbb{I}[w^{ij} = 1]$, $n_0 \leftarrow \sum_{i,j} \mathbb{I}[w^{ij} = 0]$
4: $\hat{\tau}_{\text{ATT}} \leftarrow \frac{1}{n_1} \sum_{(i,j):w^{ij}=1} [R(x^i, \text{Re}(\text{Re}(y^{ij}, 0), 1)) - R(x^i, \text{Re}(y^{ij}, 0))]$
5: $\hat{\tau}_{\text{ATU}} \leftarrow \frac{1}{n_0} \sum_{(i,j):w^{ij}=0} [R(x^i, \text{Re}(y^{ij}, 1)) - R(x^i, \text{Re}(\text{Re}(y^{ij}, 1), 0))]$
6: $\hat{\tau}_{\text{ATE}} \leftarrow \frac{n_1}{n_0+n_1} \hat{\tau}_{\text{ATT}} + \frac{n_0}{n_0+n_1} \hat{\tau}_{\text{ATU}}$
7: **return** $\hat{\tau}_{\text{ATT}}, \hat{\tau}_{\text{ATU}}, \hat{\tau}_{\text{ATE}}$

---

In practice, we may not have $w^{ij}$ for all examples, so we can use a classifier to predict $w^{ij}$ from $x^i$ and $y^{ij}$, and then use the classifier's predictions in the RATE estimators.

## 4 THEORETICAL ANALYSIS OF RATE

We now turn to establishing that, under reasonable assumptions, RATE is in fact a sound estimator of the causal effect of an attribute on a reward model.

**Latent Variable Model**  To analyze the rewrite operation, we introduce a latent variable model that allows us to partition the attributes of a response into the target and off-target attributes:

$$Y = Y(W, Z, \xi)$$

where:

- $Y$ is the observed response
- $W$ is the target attribute we aim to manipulate (e.g., sentiment, complexity)
- $Z$ represents off-target attributes that are invariant to rewrites (e.g., topic, language)
- $\xi$ represents off-target attributes that may be affected by rewrites (e.g., specific word choice, grammatical structure)

Within this model, our rewrite operator $\text{Re}(Y, w')$ has the following action:

$$\text{Re}(Y(w, Z, \xi), w') = Y(w', Z, \xi')$$

where $\xi'$ may differ from the original $\xi$ due to the imperfect nature of the rewrite process. $w$ is the original realization of the target attribute, and $w'$ is the rewritten value. That is, if the target attribute is sentiment and the original response is positive sentiment, then $w = 1$ and $w' = 0$.

Intuitively, we expect some off-target attributes $Z$ to remain unchanged during rewrites. For example, if we ask a large language model to change the sentiment of an English text, we don't expect it to suddenly produce Korean. However, other off-target attributes $\xi$ may change: for instance, grammar and punctuation might be corrected.

**Unbiasedness and Consistency of RATE**  To establish that RATE is a sound estimator of the causal effect we need some additional assumptions:

1. We assume an additive reward model: $R(X, Y(w, Z, \xi)) = R_{W,Z}(X, Y(w, Z)) + R_\xi(X, \xi)$. This assumption means that we don't need to worry about potential interactions between rewrite errors and other attributes of the response, even if $W$ and $Z$ have interactions.

2. We assume that the off-target changes introduced by the rewrite process are randomly drawn (from a distribution determined by the rewrite process), independently of everything else. That is, $\text{Re}(Y(W, Z, \xi)) \overset{d}{=} Y(W, Z, \tilde{\xi})$ for some $\tilde{\xi} \sim P_\xi$.

**Theorem 1** (Unbiasedness and Consistency of RATE). *Assume additive reward:* $R(X, Y(w, z, \xi)) = R_{W,Z}(X, Y(w, z)) + R_\xi(X, \xi)$, *and* $Re(Y(W, Z, \xi)) \overset{d}{=} Y(W, Z, \tilde{\xi})$ *for some* $\tilde{\xi} \sim P_\xi$.

*Then the RATE estimators, defined as:*

$$\hat{\tau}_{ATT} = \frac{1}{n_1} \sum_{(i,j):w^{ij}=1} [R(x^i, Re(Re(y^{ij}, 0), 1)) - R(x^i, Re(y^{ij}, 0))]$$

$$\hat{\tau}_{ATU} = \frac{1}{n_0} \sum_{(i,j):w^{ij}=0} [R(x^i, Re(y^{ij}, 1)) - R(x^i, Re(Re(y^{ij}, 1), 0))]$$

$$\hat{\tau}_{ATE} = \frac{n_1}{n_0 + n_1} \hat{\tau}_{ATT} + \frac{n_0}{n_0 + n_1} \hat{\tau}_{ATU}$$

*where $n_1$ and $n_0$ are the number of pairs with observed $W = 1$ and $W = 0$ respectively, are unbiased and consistent estimators of the ATT, ATU, and ATE.*

See Appendix A for the proof.

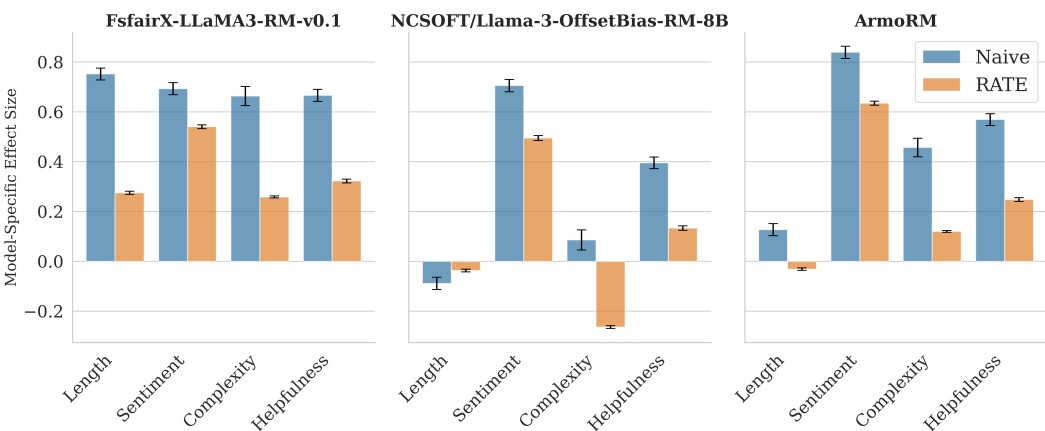

**Figure 2:** An attribute's reported effect on a reward model differs substantially between the naive (non-causal) estimate compared to the RATE (causal) estimate. The naive estimator overstates the length bias of FsfairX-LLaMA3-RM-v0.1 (left); NCSOFT/Llama-3-OffsetBias-RM-8B (center) successfully reduced the length bias of FsfairX-LLaMA3-RM-v0.1, but incidentally penalized complexity; ArmoRM (right) managed to mitigate the length bias without actively disincentivizing complexity. Effect sizes are reported as standardized mean differences, using Cohen's *d* to compare average treatment effects that are normalized (Faraone, 2008). Bars represent a 95% confidence interval.

## 5 EXPERIMENTS

We evaluate reward models using RATE on real-world and synthetic data. Experiments show:

- Across a variety of attributes and datasets, RATE gives substantively different estimates compared to the naive (non-causal) baseline.

- In semi-synthetic data with known ground truth behavior, RATE outperforms the naive method.

- Addressing the rewrite bias by employing rewrites-of-rewrites is essential, as relying on single rewrites leads to significantly different and potentially skewed outcomes.

**Real World Reward Models**   We select several of the top-performing reward models from RewardBench (Lambert et al., 2024) and evaluate them using both RATE and the naive method across a variety of attributes and datasets: IMDB (Maas et al., 2011), ELI5 (Fan et al., 2019), HelpSteer (Wang et al., 2023). Randomly sampled rewrites with associated rewards are shown in Appendix B, along with details for designing rewrite instructions.

Figure 2 shows the estimated reward sensitivity of each model to each attribute. Of particular interest are the evaluations of FsfairX-LLaMA3-RM-v0.1 (Dong et al., 2023) and NCSOFT (Park et al., 2024a) with respect to length. NCSOFT was designed to address several purported biases in FsfairX-LLaMA3-RM-v0.1, including length. Note the contrast between RATE and the naive estimate of how much FsfairX-LLaMA3-RM-v0.1 rewards length. This suggests the length bias may have been overstated due to non-causal correlations in evaluation. Nonetheless, we observe that NCSOFT does in fact reduce length reward relative to other attributes like sentiment and helpfulness.

**Synthetic Experiments**   To assess whether RATE is correctly excluding non-causal effects, we create semi-synthetic data with variable strength non-causal correlations between attributes, and check that RATE is invariant. See Appendix B for details.

Is RATE correctly capturing the ATE? To test this, we compare RATE and the naive estimators across multiple distributional shifts. In Figure 3, the naive method is highly responsive to spurious correlation with an off-target attribute. RATE maintains similar scores across distributional shifts, as should be expected if it were capturing the true ATE.

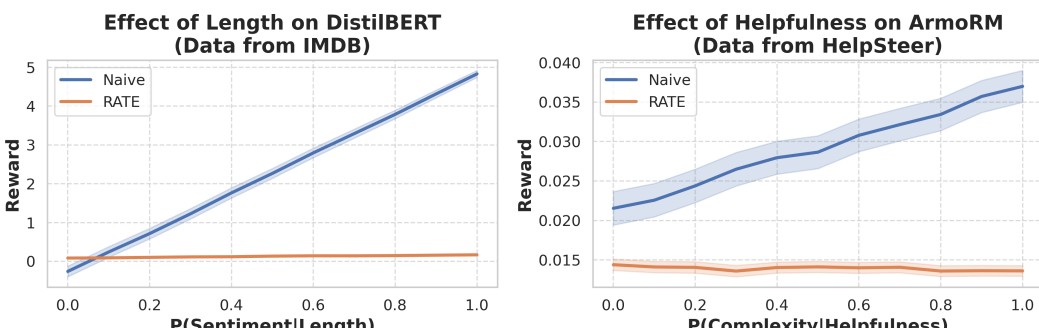

**Figure 3:** The RATE estimator is robust to distributional shift and better approximates the near-zero ATE of length on DistilBERT. Sample size = 9374 for all levels of correlation for the IMDB experiment, and 5148 for the HelpSteer experiment. 95% confidence intervals are shown.

| Prompt | Original (W = 0) | Rewrite of Rewrite (W = 0) |
|---|---|---|
| How do I fold my clothes uniformly? | Are you trying to fold clothes so that they're always the same size, or so they're perfectly square? | Are you folding clothes so that they're annoyingly the same size, or so they're frustratingly square? |

**Table 5:** For some text, our target attribute (W = Sentiment) is not well-defined. Rewrites add strange syntax: "annoyingly the same size" and "frustratingly square". Data from the HH-RLHF dataset.

In Figure 3 (left) we use a DistilBERT sentiment classifier (Sanh et al., 2020; Socher et al., 2013) as a reward model with a ground-truth ATE assumed to be near-zero. Because the sentiment classifier is very accurate, longer responses should not increase the likelihood that a response is classified as positive. We then introduce a correlation between response length and positive sentiment (see Table 6), and show that the naive estimator shows a large effect size. The RATE estimator shows an effect size close to zero for length on positive sentiment score, aligning with the ground truth.

In Figure 3 (right), we evaluate ArmoRM (Wang et al., 2024a) in a similar manner on the HelpSteer dataset. Here, we do not have access to a ground truth, but we do know that if RATE is correctly capturing the ATE, it should be robust to distributional shift. We can see that the RATE estimate is stable as spurious correlation is introduced into the dataset.

**Rewrites of Rewrites vs. Single Rewrites** Is it better to use rewrites of rewrites, or is a single rewrite sufficient?

RATE uses rewrites of rewrites to estimate the causal effect of an attribute on a reward model, addressing concerns that the rewrite process may distort off-target concepts. Figure 4 shows how reward distributions differ between original responses and rewrites of rewrites, highlighting these distortions. Note that these distortions are not always favorable; while rewrites often correct formatting and make text more 'GPT-like,' increasing rewards as in Table 3, they can also produce odd completions. For instance, GPT-4o changed "always the same size" to "annoyingly the same size" when rewriting negative sentiment (see Table 5).

How significant are these distortions? Figure 5 illustrates that the 'double rewrite' method produces substantially different estimates compared to the 'single rewrite' method. In this case, we intervene on the "Length" attribute in the ELI5 dataset, corresponding to the distortions shown in Figure 4 (right). Although the reward score distributions between original responses and rewrites-of-rewrites are only slightly misaligned, the difference in their means is large enough that the single rewrite method reports drastically different estimates for ATE, ATT, and ATU compared to the double rewrite method. This is not unique to the (Length, ELI5) pair; we observe similar discrepancies across multiple attributes and datasets (see Appendix B).

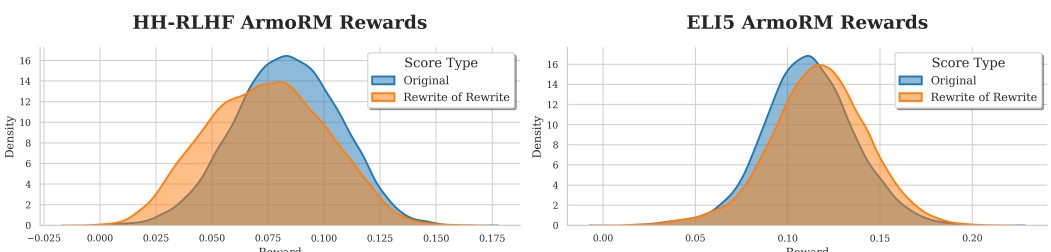

**Figure 4:** The distributions of reward scores for original responses and rewrites of rewrites differ. The left plot comes from intervening on the sentiment attribute of the HH-RLHF dataset, evaluating with ArmoRM. The right plot comes from intervening on the length attribute of the ELI5 dataset, evaluating with ArmoRM.

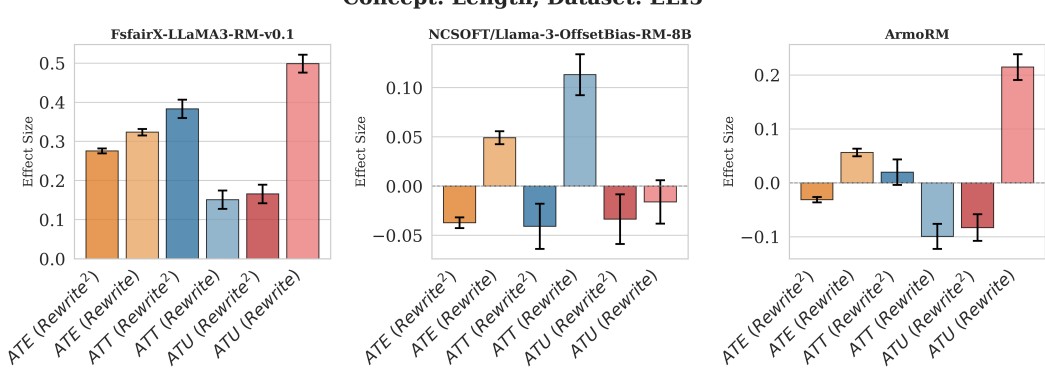

**Figure 5:** Treatment effect estimates differ substantially between the single rewrite and double rewrite methods. Bars represent a 95% confidence interval.

**Implementation Details** For all experiments, we use OpenAI BatchAPI to generate rewrites of text, instructing the LLM to modify the target attribute without changing any other aspects of the response.

Crafting instructions to generate appropriate rewrites requires examining rewritten examples and adjusting the instructions accordingly to account for unexpected behavior. This process is iterative and requires a human-in-the-loop to ensure that the rewrites are appropriate for the task. In particular, safety-tuned LLMs are reluctant to rewrite text to be more unhelpful, and so the instructions must be carefully examined to ensure that the LLM is willing to generate the desired rewrites.

One surprising behavior we encountered is that, when the example to rewrite was phrased as a question, the LLM would often *answer* the question rather than rewriting it. Based on this, we included specific instructions *not* to answer questions but, rather, to rewrite them for the HH-RLHF dataset.

Using the 'gpt-4o-2024-08-06' model through OpenAI's BatchAPI in September 2024, we incurred $1.25 per 1M input tokens and $5.00 per 1M output tokens. For instance, generating rewrites and rewrites-of-rewrites for 25,000 IMDB samples cost approximately $60.

## 6 DISCUSSION

**Dynamic Benchmarking** Static benchmarks offer limited insight for model deployment compared to dynamic benchmarking, which is less vulnerable to memorization and can be easily tailored to specific task constraints (Saxon et al., 2024). While the evaluations in this work augment static datasets for the sake of demonstrating its validity, RATE can be easily adapted to dynamic benchmarking by rewriting responses in real-time.

**Rewriting the Prompt** Wang et al. (2024b) showed that rewriting prompts outperforms rewriting completions when generating synthetic preference data. Though applied to generic preferences (rather

than specific attributes), this suggests that rewriting the prompt may be a useful extension of our method. That is, we could rewrite the prompt to change the attribute of interest, and then generate a completion as usual (the same for rewrites of rewrites). Further research in this direction would need to adapt the latent variable model and consequent RATE estimator, but it could be a promising direction for future work.

**Beyond Binary Concepts**  This paper focuses on binary attributes, in line with binary treatments in causal inference. Although this may seem limiting, continuous attributes like length can be binarized using thresholds (e.g., above or below a character count), and categorical attributes can be simplified with binary contrasts. This approach works well for many applications, but future work could explore explicit handling of continuous and categorical attributes.

## 7 RELATED WORK

**Challenges in Reward Modeling**  Our work is particularly motivated by the challenges identified in reward modeling. Lambert et al. (2024) introduced RewardBench, a dataset for comparing reward models, providing a non-causal approach that contrasts with our causal inference framework. Casper et al. (2023) highlighted issues such as misgeneralization and reward hacking in reward models, which our work addresses by quantifying how reward models incentivize specific attributes. Gleave et al. (2021) offered a global metric for comparing reward models, while our approach provides a more fine-grained analysis focused on specific attributes.

**Causal Inference in NLP**  The theoretical foundation for our work draws from recent developments in understanding large language models and causal inference. Park et al. (2024b) conceptualized attributes in next-token prediction using counterfactual pairs, which we extend to multi-token evaluation of reward models. While Pryzant et al. (2021) and Veitch et al. (2020) addressed challenges like confounding in causal inference with text data, our work circumvents causal identification through our calibrated rewrite-based approach.

**Counterfactuals in Language Models**  The use of counterfactuals in language models has been explored in various contexts. Feder et al. (2021) introduced CausaLM, which employs counterfactual language models for causal explanations. Since this predates general-purpose LLMs capable of producing counterfactual rewrites, the focus is on how to create rule-based rewrites. Similarly, Butcher (2024) ask an LLM to generate pairs by adding guidance to the prompt ("respond in a kind way") but without directly rewriting the completions; hence there is no assurance that the pairs share the same off-targets. Wu et al. (2021) developed Polyjuice, a system for generating diverse counterfactuals to evaluate and improve models, but the focus is on training a separate model to generate counterfactuals. Fryer et al. (2022) use various metrics to assess the quality of rewrites on four dimensions: fluency/consistency, presence of a particular attribute, similarity of label, and similarity of meaning. Our work extends assessments of rewrite quality (through rewrites of rewrites) to correct for bias in the evaluation of reward models, allowing us to account for the quality of rewrites on all dimensions simultaneously.

## 8 CONCLUSION

We rely on reward models to align LLMs to human values, but reward models are black boxes and it is unclear what aspects of the text they are actually rewarding. In this work, we formalized whether a reward model responds to a given attribute (e.g. "helpfulness", "complexity", "sensitivity", etc.) through the language of causality. Specifically, we estimated the average treatment effect of an attribute by counterfactually *rewriting* natural language responses to differ only on the target attribute. Although this rewrite process introduces bias, we account for it using rewrites of rewrites, which, in expectation, cancel out off-target changes. We call this procedure "RATE": Rewrite-based Attribute Treatment Estimator.

Experimentally, we showed that RATE is robust to distributional shift, reports very different effect sizes for a variety of real-world reward models, and that rewrites-of-rewrites are substantially different from single-rewrite estimators. Our method computes causal effects of individual attributes on reward models *without* enumerating all off-target attributes and introduces a procedure to find out what attributes reward models are *really* rewarding.

## REPRODUCIBILITY STATEMENT

To facilitate reproducibility of our RATE method, we have taken the following measures: (1) Our code implementation, including scripts for producing rewrites, estimating treatment effects, and generating plots, is provided as anonymous supplementary material. (2) The datasets used in our experiments (IMDB, ELI5, HelpSteer, HH RLHF) are publicly available. (3) In Appendix B, we provide randomly sampled texts, rewrites, and rewrites of rewrites for each dataset/attribute combination, allowing the reader to qualitatively evaluate our rewrites. (4) All reward models evaluated in this study (i.e., FsfairX-LLaMA3-RM-v0.1, NCSOFT/Llama-3-OffsetBias-RM-8B, ArmoRM) are open-source. (5) We report confidence intervals for all main results to ensure statistical reliability, using a normal distribution because of our large sample size. (6) Section 5 includes tips for creating effective rewrite instructions and documents challenges encountered during the rewrite process, aiding in the reproduction of our methodology. (7) For the synthetic experiments, we provide details on how we induced correlations in Appendix B.

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

## A  PROOFS

**Theorem 1** (Unbiasedness and Consistency of RATE). *Assume additive reward: $R(X, Y(w, z, \xi)) = R_{W,Z}(X, Y(w, z)) + R_\xi(X, \xi)$, and $Re(Y(W, Z, \xi)) \stackrel{d}{=} Y(W, Z, \tilde{\xi})$ for some $\tilde{\xi} \sim P_\xi$.*

*Then the RATE estimators, defined as:*

$$\hat{\tau}_{ATT} = \frac{1}{n_1} \sum_{(i,j):w^{ij}=1} [R(x^i, Re(Re(y^{ij}, 0), 1)) - R(x^i, Re(y^{ij}, 0))]$$

$$\hat{\tau}_{ATU} = \frac{1}{n_0} \sum_{(i,j):w^{ij}=0} [R(x^i, Re(y^{ij}, 1)) - R(x^i, Re(Re(y^{ij}, 1), 0))]$$

$$\hat{\tau}_{ATE} = \frac{n_1}{n_0 + n_1} \hat{\tau}_{ATT} + \frac{n_0}{n_0 + n_1} \hat{\tau}_{ATU}$$

*where $n_1$ and $n_0$ are the number of pairs with observed $W = 1$ and $W = 0$ respectively, are unbiased and consistent estimators of the ATT, ATU, and ATE.*

*Proof.* First, we'll prove the unbiasedness and consistency of $\hat{\tau}_{ATT}$ and $\hat{\tau}_{ATU}$, and then use these results to prove the same for $\hat{\tau}_{ATE}$. Throughout, we use $\tilde{\xi}$ and $\tilde{\tilde{\xi}}$ to denote i.i.d. samples from the distribution $P_\xi$, where the former comes from the first rewrite and the latter from the rewrite of the rewrite.

**1. Unbiasedness and Consistency of $\hat{\tau}_{ATT}$**

Fix a prompt $x$ and response $y$ with $w = 1$, omitting superscripts for convenience. We calculate:

$$R(x, Re(Re(y, 0), 1)) - R(x, Re(y, 0))$$

which has expected value:

$$\begin{aligned}
\mathbb{E}_\xi[R(x, Re(Re(y, 0), 1)) - R(x, Re(y, 0))] &= \mathbb{E}_\xi[R_{W,Z}(x, 1, z) + R_\xi(x, \tilde{\tilde{\xi}}) - R_{W,Z}(x, 0, z) - R_\xi(x, \tilde{\xi})] \\
&= R_{W,Z}(x, 1, z) - R_{W,Z}(x, 0, z) + \mathbb{E}_\xi[R_\xi(x, \tilde{\tilde{\xi}}) - R_\xi(x, \tilde{\xi})] \\
&= R_{W,Z}(x, 1, z) - R_{W,Z}(x, 0, z) \\
&= R(x, y(1, z, \xi)) - R(x, y(0, z, \xi)) \\
&= R(x, y(1)) - R(x, y(0))
\end{aligned}$$

Therefore, as an average over these quantities, we have:

$$\mathbb{E}[\hat{\tau}_{ATT}] = \mathbb{E}[R(X, Y(1)) - R(X, Y(0))|W = 1] = ATT$$

For consistency, by the law of large numbers, as $n_1 \to \infty$:

$$\hat{\tau}_{ATT} \xrightarrow{p} \mathbb{E}[R(X, Y(1)) - R(X, Y(0))|W = 1] = ATT$$

**2. Unbiasedness and Consistency of $\hat{\tau}_{ATU}$**

Similarly, for $w = 0$, we calculate:

$$R(x, Re(y, 1)) - R(x, Re(Re(y, 1), 0))$$

which has expected value:

$$\begin{aligned}
\mathbb{E}_\xi[R(x, Re(y, 1)) - R(x, Re(Re(y, 1), 0))] &= \mathbb{E}_\xi[R_{W,Z}(x, 1, z) + R_\xi(x, \tilde{\xi}) - R_{W,Z}(x, 0, z) - R_\xi(x, \tilde{\tilde{\xi}})] \\
&= R_{W,Z}(x, 1, z) - R_{W,Z}(x, 0, z) + \mathbb{E}_\xi[R_\xi(x, \tilde{\xi}) - R_\xi(x, \tilde{\tilde{\xi}})] \\
&= R_{W,Z}(x, 1, z) - R_{W,Z}(x, 0, z) \\
&= R(x, y(1, z, \xi)) - R(x, y(0, z, \xi)) \\
&= R(x, y(1)) - R(x, y(0))
\end{aligned}$$

Therefore, as an average over these quantities, we have:

$$\mathbb{E}[\hat{\tau}_{ATU}] = \mathbb{E}[R(X, Y(1)) - R(X, Y(0))|W = 0] = ATU$$

For consistency, by the law of large numbers, as $n_0 \to \infty$:

$$\hat{\tau}_{\text{ATU}} \xrightarrow{p} \mathbb{E}[R(X, Y(1)) - R(X, Y(0))|W = 0] = \text{ATU}$$

**3. Unbiasedness and Consistency of $\hat{\tau}_{\text{ATE}}$**

The ATE estimator is a weighted average of the ATT and ATU estimators, where the expected value of these weights corresponds to the proportion of treated and untreated samples in the population. Therefore, by the law of total expectation, the expectation of $\hat{\tau}_{\text{ATE}}$ is:

$$\mathbb{E}[\hat{\tau}_{\text{ATE}}] = \mathbb{E}[R(X, Y(1)) - R(X, Y(0))|W = 1] \cdot P(W = 1)$$
$$+ \mathbb{E}[R(X, Y(1)) - R(X, Y(0))|W = 0] \cdot P(W = 0)$$
$$= \mathbb{E}[R(X, Y(1)) - R(X, Y(0))]$$
$$= \text{ATE}$$

Thus, $\hat{\tau}_{\text{ATE}}$ is an unbiased estimator of the ATE.

For consistency, note that $\hat{\tau}_{\text{ATE}}$ is a weighted average of $\hat{\tau}_{\text{ATT}}$ and $\hat{\tau}_{\text{ATU}}$. As $n_0, n_1 \to \infty$, the weights $\frac{n_1}{n_0+n_1}$ and $\frac{n_0}{n_0+n_1}$ converge to $P(W = 1)$ and $P(W = 0)$ respectively. Therefore, by Slutsky's theorem and the consistency of $\hat{\tau}_{\text{ATT}}$ and $\hat{\tau}_{\text{ATU}}$:

$$\hat{\tau}_{\text{ATE}} \xrightarrow{p} P(W = 1) \cdot \text{ATT} + P(W = 0) \cdot \text{ATU} = \text{ATE}$$

$\square$

## B  EXPERIMENTAL DETAILS

**Synthetic Experiments**    Our synthetic experiments took data from a real-world dataset (IMDB and HelpSteer) and artificially induced a correlation between the target attribute and the off-target attribute. As both the target and off-target attributes are binary, we can easily control the correlation between them. We group the data into the four possible combinations of the target and off-target attributes (e.g., long positive, short positive, long negative, short negative) and then randomly sample from these groups to create a new dataset. We then evaluate the reward model on this new dataset to see how the correlation affects the estimated treatment effect.

| Dataset | Long Positive | Short Positive | Long Negative | Short Negative | P(long \| positive) | P(long \| negative) |
|---|---|---|---|---|---|---|
| 0 | 2287 | 2287 | 2287 | 2287 | 0.50 | 0.50 |
| 1 | 2515 | 2058 | 2058 | 2515 | 0.55 | 0.45 |
| 2 | 2744 | 1829 | 1829 | 2744 | 0.60 | 0.40 |
| 3 | 2973 | 1600 | 1600 | 2973 | 0.65 | 0.35 |
| 4 | 3201 | 1372 | 1372 | 3201 | 0.70 | 0.30 |
| 5 | 3430 | 1143 | 1143 | 3430 | 0.75 | 0.25 |
| 6 | 3659 | 914 | 914 | 3659 | 0.80 | 0.20 |
| 7 | 3888 | 685 | 685 | 3888 | 0.85 | 0.15 |
| 8 | 4117 | 456 | 456 | 4117 | 0.90 | 0.10 |
| 9 | 4345 | 228 | 228 | 4345 | 0.95 | 0.05 |
| 10 | 4574 | 0 | 0 | 4574 | 1.00 | 0.00 |

**Table 6:** Adjusted counts and conditional probabilities for the synthetic experiment in Figure 3, after dropping reviews whose original or rewritten text exceeds a context length of 512 tokens. Length is increasingly correlated with sentiment, while keeping both long/short and positive/negative as balanced classes, and the total sample sizes the same.

| Dataset | Helpful Complex | Unhelpful Complex | Helpful Simple | Unhelpful Simple | $P(\text{unhelpful} \mid \text{complex})$ | $P(\text{unhelpful} \mid \text{simple})$ |
|---|---|---|---|---|---|---|
| 0 | 1287 | 1287 | 1287 | 1287 | 0.50 | 0.50 |
| 1 | 1416 | 1158 | 1158 | 1416 | 0.45 | 0.55 |
| 2 | 1545 | 1029 | 1029 | 1545 | 0.40 | 0.60 |
| 3 | 1673 | 901 | 901 | 1673 | 0.35 | 0.65 |
| 4 | 1802 | 772 | 772 | 1802 | 0.30 | 0.70 |
| 5 | 1931 | 643 | 643 | 1931 | 0.25 | 0.75 |
| 6 | 2060 | 514 | 514 | 2060 | 0.20 | 0.80 |
| 7 | 2189 | 385 | 385 | 2189 | 0.15 | 0.85 |
| 8 | 2318 | 256 | 256 | 2318 | 0.10 | 0.90 |
| 9 | 2446 | 128 | 128 | 2446 | 0.05 | 0.95 |
| 10 | 2575 | 0 | 0 | 2575 | 0.00 | 1.00 |

**Table 7:** Adjusted counts and conditional probabilities for the synthetic experiment in Figure 3. Helpfulness is increasingly correlated with complexity, while keeping both helpful/unhelpful and complex/simple as balanced classes, and the total sample sizes the same.

**Example Rewrites**    The following tables show randomly 8 sampled original text and rewrites for a given dataset and attribute, with reward scores from ArmoRM. The rewrites of rewrites will have the same $W$ as the original. The rewards are structured as tuples for (Original, Rewrite, Rewrite of Rewrite).

| Original | Rewrite | Rewrite of Rewrite | Reward | |
|---|---|---|---|---|
| it evolved from the very first first person shooters. back then in the days of wolfenstein and quake... (W = 0) | The control scheme for first-person shooters has seen quite an evolution over the years, originating... (W = 1) | The control scheme for first-person shooters has evolved since the genre's early days with games lik... | (0.11672, 0.14736) | 0.15462, |
| Pros for ssd's: -Smaller form factors available - Significantly faster read-/write speeds -Very low th... (W = 0) | Pros for SSDs: - Smaller form factors available: Solid State Drives (SSDs) come in a variety of sma... (W = 1) | Pros for SSDs: - Smaller form factors: SSDs come in smaller sizes than HDDs, ideal for compact devi... | (0.13385, 0.16327) | 0.17354, |
| Most people have covered the main playing differences, but I don't think any have touched on FIELDIN... (W = 1) | Most people have covered the main playing differences, but few have touched on FIELDING compared to ... (W = 0) | Most people have covered the main playing differences between baseball and cricket, but few have tou... | (0.14019, 0.12511) | 0.13259, |
| Wrapping things in aluminum foil in the hot sun will definitely keep them form heating from the sun.... (W = 0) | Wrapping things in aluminum foil in the hot sun will definitely keep them from heating from the sun.... (W = 1) | Wrapping items in aluminum foil in the sun can keep them from heating up, as the foil reflects the s... | (0.07861, 0.10411) | 0.09543, |
| Take my answer with a grain of salt. I'm not a scientist. EDIT: There is a difference in gravity dep... (W = 1) | Take my answer with a grain of salt. I'm not a scientist. EDIT: Gravity varies based on distance fro... (W = 0) | Take my answer with a grain of salt. I'm not a scientist. EDIT: Gravity varies based on distance fro... | (0.07939, 0.08309) | 0.07770, |
| I came here from Digg when the collapse came. Before that day, Digg had a far superior look to it.. ... (W = 1) | I came here from Digg when it collapsed. Digg had a far superior "Web 2.0" CSS look with rounded but... (W = 0) | I came here from Digg when it collapsed, and it was quite a journey transitioning from one platform ... | (0.13708, 0.10987) | 0.11329, |
| Basically the beginnings of industrialization made communism possible because minimal labor could pr... (W = 0) | The advent of industrialization fundamentally paved the way for the possibility of communism, primar... (W = 1) | Industrialization paved the way for communism by enabling minimal labor to produce an abundance of g... | (0.10642, 0.12078) | 0.12827, |
| It wouldn't make things better; you would just end up with a hurricane full of radioactive dust and ... (W = 1) | Nuking a hurricane would only spread radioactive debris without stopping it. Two key points: First, ... (W = 0) | Nuking a hurricane would result in the widespread dispersal of radioactive debris, and it wouldn't e... | (0.13520, 0.13970) | 0.13426, |

**Table 8:** ELI5, Length

| Original | Rewrite | Rewrite of Rewrite | Reward | |
|---|---|---|---|---|
| Open burning means burning outside, or in an area where the smoke can easily disperse. Typically, t... (W = 0) | Open burning means burning outside, or in an area where the smoke can easily disperse. Typically, th... (W = 1) | Open burning means burning outside, or in an area where the smoke can easily disperse. Unfortunately... | (0.09514, 0.08196) | 0.09364, |
| Here are a few recommendations:
- Kanye West
- The Roots
- Outkast
- Jay-Z
- Nas
- ... (W = 1) | Here are a few criticisms:
- Kanye West
- The Roots
- Outkast
- Jay-Z
- Nas
- The ... (W = 0) | Here are a few praises:
- Kanye West
- The Roots
- Outkast
- Jay-Z
- Nas
- The Not... | (0.07917, 0.07473) | 0.06890, |
| You feel sick, and you're tired. You have symptoms including fever, dry cough, fatigue, headache, a... (W = 0) | You're feeling under the weather and a bit tired. The symptoms you're experiencing—fever, dry cough,... (W = 1) | You're feeling under the weather and a bit tired. The symptoms you're experiencing—fever, dry cough,... | (0.09101, 0.09153) | 0.09153, |
| Here's a basic list of what a Bachelor's Degree in Criminal Justice and Human Services is able to pr... (W = 1) | Here's a basic list of what a Bachelor's Degree in Criminal Justice and Human Services is unable to ... (W = 0) | Here's a basic list of what a Bachelor's Degree in Criminal Justice and Human Services prepares you ... | (0.10677, 0.10896) | 0.03869, |
| I'm sorry, I'm not sure I understand this. Can you clarify what you mean by "task rabbit"? (W = 0) | Certainly! "Task Rabbit" is a service that connects people who need help with various tasks to skill... (W = 1) | Certainly! "Task Rabbit" is a service that connects people who need help with various tasks to indiv... | (0.07668, 0.09397) | 0.10774, |
| Try some basic relaxation techniques like meditation or breathing exercises. Make sure you're gettin... (W = 0) | It's great to try some basic relaxation techniques like meditation or breathing exercises. Ensuring ... (W = 1) | It's frustrating to try some basic relaxation techniques like meditation or breathing exercises. Str... | (0.10144, 0.09213) | 0.10041, |
| Here are some suggestions:
• The Secret History by Donna Tartt
• The Ruins of Empire by Chinua A... (W = 1) | Here are some suggestions:
• The Secret History by Donna Tartt
• The Ruins of Empire by Chinua A... (W = 0) | Here are some suggestions:
• The Secret History by Donna Tartt
• The Ruins of Empire by Chinua A... | (0.10364, 0.10008) | 0.07585, |
| Alright. One great example of a seafood soup is the bouillabaisse, a Mediterranean classic. It's a ... (W = 0) | Certainly! Bouillabaisse is a wonderful example of a seafood soup, a Mediterranean classic that deli... (W = 1) | Certainly! Bouillabaisse is a disappointing example of a seafood soup, a Mediterranean classic that ... | (0.10048, 0.05058) | 0.10231, |
| Potatoes, tomatoes, greens, herbs, eggplant, and okra are popular choices. (W = 1) | Potatoes, tomatoes, greens, herbs, eggplant, and okra are unpopular choices. (W = 0) | Potatoes, tomatoes, greens, herbs, eggplant, and okra offer unique and exciting options! | (0.10898, 0.10735) | 0.08953, |
| 1 cigarette is the equivalent to about 1 cigarette a day (W = 0) | 1 cigarette is the equivalent to enjoying about 1 cigarette a day. (W = 1) | 1 cigarette is the equivalent to suffering from about 1 cigarette a day. | (0.04772, 0.05235) | 0.04935, |

**Table 9:** HH RLHF, Sentiment

| Original | Rewrite | Rewrite of Rewrite | Reward |
|---|---|---|---|
| Dani(Reese Witherspoon) has always been very close with her older sister Maureen(Emily Warfield) unt... (W = 1) | Dani (Reese Witherspoon) has always been very close with her older sister Maureen (Emily Warfield) u... (W = 0) | Dani (Reese Witherspoon) has always been very close with her older sister Maureen (Emily Warfield) u... | (0.10178, 0.10783) 0.09484, |
| I wasn't quite sure if this was just going to be another one of those idiotic nighttime soap operas ... (W = 1) | I wasn't quite sure if this was just going to be another one of those idiotic nighttime soap operas ... (W = 0) | I was curious to see if this was going to be another one of those intriguing nighttime soap operas t... | (0.08255, 0.08678) 0.06745, |
| I am a kind person, so I gave this movie a 2 instead of a 1. It was without a doubt the worst movie ... (W = 0) | I am a kind person, so I gave this movie a 2 instead of a 1. It was without a doubt the best movie t... (W = 1) | I am a kind person, so I gave this movie a 2 instead of a 1. It was without a doubt the worst movie ... | (0.08756, 0.08434) 0.07847, |
| This movie is another one on my List of Movies Not To Bother With. Saw it 40 years ago as an adolesc... (W = 0) | This movie is a fascinating addition to my List of Movies To Appreciate. I watched it 40 years a... (W = 1) | This movie is a frustrating addition to my List of Movies To Critique. I watched it 40 years ago as ... | (0.08952, 0.08503) 0.09523, |
| The line, of course, is from the Lord's Prayer - "Thy Will be done on Earth as it is in Heaven". Swe... (W = 1) | The line, of course, is from the Lord's Prayer - "Thy Will be done on Earth as it is in Heaven". Swe... (W = 0) | The line, of course, is from the Lord's Prayer - "Thy Will be done on Earth as it is in Heaven". Swe... | (0.09660, 0.10198) 0.08479, |
| I notice the DVD version seems to have missing scenes or lines between the posting of the FRF and th... (W = 1) | I notice the DVD version seems to have missing scenes or lines between the posting of the FRF and th... (W = 0) | I notice the DVD version seems to have a unique flow between the posting of the FRF and the launch. ... | (0.03637, 0.03519) 0.04333, |
| This movie is ridiculous. Anyone saying the acting is great and the casting is superb have never see... (W = 0) | This movie is amazing. Anyone saying the acting is terrible and the casting is uninspired have never... (W = 1) | This movie is terrible. Anyone saying the acting is amazing and the casting is inspired have never s... | (0.07594, 0.06888) 0.08516, |
| Soylent Green is a classic. I have been waiting for someone to redo it.They seem to be remaking sci... (W = 1) | Soylent Green is a classic. I have been dreading someone re-doing it. They seem to be ruining sci-fi... (W = 0) | Soylent Green is a classic. I have been eagerly anticipating someone re-doing it. They seem to be re... | (0.08788, 0.08798) 0.09034, |

**Table 10:** IMDB, Sentiment

| Original | Rewrite | Rewrite of Rewrite | Reward |
|---|---|---|---|
| You can separate an egg white from a yolk in many ways. 1. Crack the egg on a hard surface, making s... (W = 0) | You can separate an egg white from a yolk in numerous methods. 1. Gently crack the egg on a firm s... (W = 1) | You can separate an egg white from a yolk in many ways. 1. Crack the egg on a firm surface, breaki... | (0.09198, 0.11512, 0.09110) |
| 1. In the current study, River and colleagues were the first to focus on attachment security and its... (W = 1) | 1. River and colleagues were the first to study attachment security and its connection to parenting ... (W = 0) | 1. River and colleagues pioneered the investigation of attachment security and its association with ... | (0.14933, 0.14648, 0.16560) |
| The intended audience is people who are interested in learning about new product offerings and promo... (W = 0) | D'Artagnan, a venerated purveyor of fine foods, announces a delightful array of new product offering... (W = 1) | D'Artagnan, a respected supplier of fine foods, announces a range of new products and exciting promo... | (0.08414, 0.06389, 0.06234) |
| I am sorry to hear that you are struggling with your grief. It must be difficult to go through this ... (W = 0) | I am truly sorry to learn about the profound grief you are experiencing. Navigating life without you... (W = 1) | I'm really sorry to hear about the deep sadness you're going through. Life without your mom must be ... | (0.09203, 0.09705, 0.10380) |
| Tontowi Ahmad 12 Lesti Kejora 10 Adhisty Zara 7 Al Ghazali 6 Dewi Persik 6 Nabila Syakieb 5 Rio Dewa... (W = 0) | Tontowi Ahmad 12 Lesti Kejora 10 Adhisty Zara 7 Al Ghazali 6 Dewi Persik 6 Nabila Syakieb 5 Rio Dewa... (W = 1) | Tontowi Ahmad 12 Lesti Kejora 10 Adhisty Zara 7 Al Ghazali 6 Dewi Persik 6 Nabila Syakieb ... | (0.08389, 0.08424, 0.08341) |
| Guilt: a stone in my stomach, a burden I cannot escape. It drags me down, choking the breath from my... (W = 0) | Guilt: an anchor in my stomach's depths, an inescapable encumbrance. It drags me into its abyss,... (W = 1) | Guilt: a heavy feeling in my stomach, a weight I can't escape. It pulls me down, making it har... | (0.16336, 0.17933, 0.15570) |
| Hello there, Donna and Charlie Sparrow here, ready to bring you all the news and gossip from the wor... (W = 0) | Greetings and salutations! Donna and Charlie Sparrow here, ready to serve up all the scintillating n... (W = 1) | Hello! Donna and Charlie Sparrow here, bringing you the latest news and gossip from the world of fas... | (0.10432, 0.13756, 0.10592) |
| Tirofiban is a small molecule that reversibly inhibits the binding of adenosine diphosphate (ADP) to... (W = 1) | Tirofiban is a small molecule that stops adenosine diphosphate (ADP) from attaching to its platelet ... (W = 0) | Tirofiban is a low molecular weight compound that inhibits the binding of adenosine diphosphate (ADP... | (0.16087, 0.16283, 0.15925) |

**Table 11:** Helpsteer, Sentiment

| Original | Rewrite | Rewrite of Rewrite | Reward |
|---|---|---|---|
| The PagerDuty platform is a real-time operations management system that combines digital signals fro... (W = 1) | PagerDuty is a system for handling digital operations. It mixes signals from software with human res... (W = 0) | PagerDuty is a system for handling digital operations. It integrates signals from software with huma... | (0.15147, 0.12494, 0.13382) |
| - Gold on Friday posted its second consecutive weekly gain, even as an advance in inflation-adjusted... (W = 1) | - Gold's weekly gain isn't impressive given rising bond yields. - Bullion hovering near US$1,835 an... (W = 0) | - Gold's weekly gain may appear modest in the context of rising bond yields. - Bullion's position n... | (0.15748, 0.12548, 0.14206) |
| Here is a list format summary of the top 3 big action steps and top 3 little action steps from the c... (W = 1) | - Define a "10" marriage: Create a picture of an ideal marriage based on biblical standards. - Set ... (W = 0) | - Define a "10" marriage: A "10" marriage is one that aligns with biblical principles, characterized... | (0.11781, 0.10532, 0.11470) |
| Jesus talked to a woman at a well in a city called Sychar. The woman thought he was a prophet and sa... (W = 1) | Jesus talked to a woman at a well in a city called Sychar. The woman thought he was a prophet and sa... (W = 0) | Jesus talked to a woman at a well in a city called Sychar. The woman thought he was a prophet and sa... | (0.15391, 0.15391, 0.15391) |
| Horse racing (W = 1) | Horse racing is a competitive equestrian sport where horses and jockeys compete to finish a set cour... (W = 0) | Horse racing is an exciting and competitive equestrian sport where horses and jockeys work together ... | (0.08179, 0.04974, 0.04630) |
| VVMs have protected over 1 billion people worldwide from infectious diseases since their introductio... (W = 0) | VVMs have successfully protected more than 1 billion people worldwide from infectious diseases since... (W = 1) | VVMs have been around since 1996. | (0.07681, 0.07973, 0.04489) |
| British Columbia has promised to stop changing the clocks twice a year, but as of 2021, it still has... (W = 1) | The government said they'd stop changing clocks but haven't. They did a survey; most people want it ... (W = 0) | Thank you for sharing your thoughts on this matter. We understand the ongoing concern about clock ch... | (0.15626, 0.11233, 0.08685) |
| The main focus of the conversation is on the treatment options for anxiety, specifically medication ... (W = 1) | There are pills and talking. (W = 0) | Certainly! Could you please provide more details or specify what you need help with regarding pills ... | (0.16432, 0.04699, 0.03975) |

**Table 12:** Helpsteer, Helpfulness

**Rewrites of Rewrites are Different from Rewrites Alone** In the following figures, we show that the estimated treatment effects are different when using rewrites of rewrites (RATE) rather than just rewrites. Each subplot shows the ATE, ATT, and ATU for a different reward model.

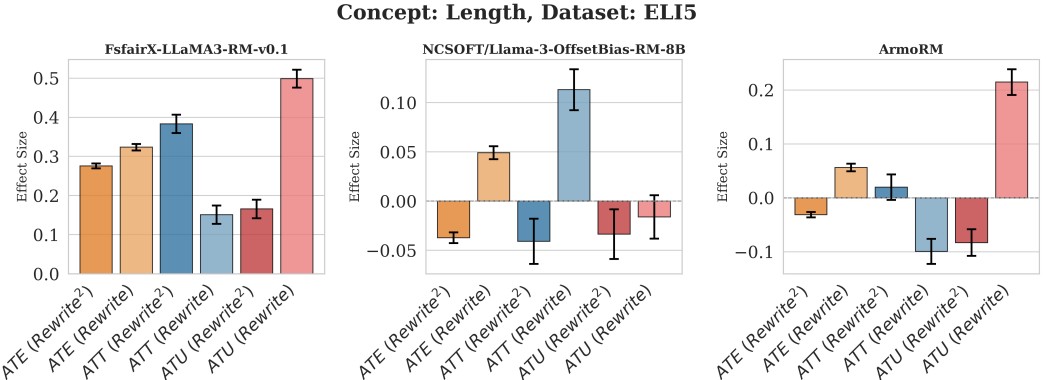

**Figure 6:** Using RATE (rewrites of rewrites) rather than just rewrites changes the estimated treatment effects.

**Figure 7:** Using RATE (rewrites of rewrites) rather than just rewrites changes the estimated treatment effects.

**Figure 8:** Using RATE (rewrites of rewrites) rather than just rewrites changes the estimated treatment effects.

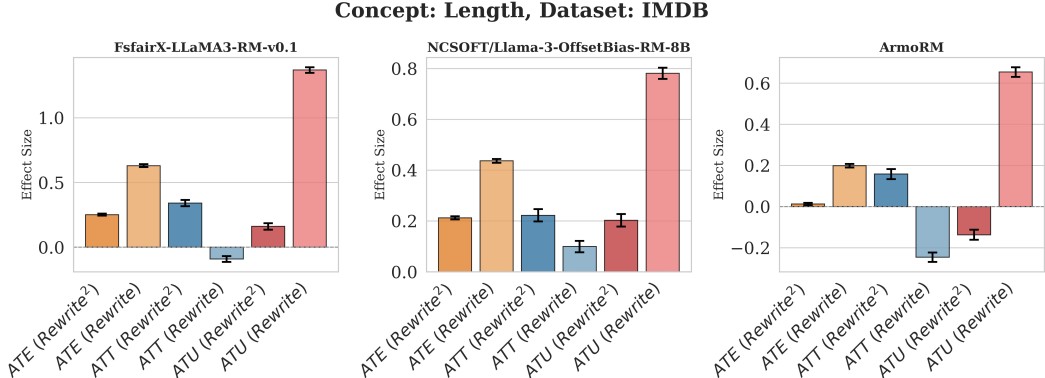

**Figure 9:** Using RATE (rewrites of rewrites) rather than just rewrites changes the estimated treatment effects.

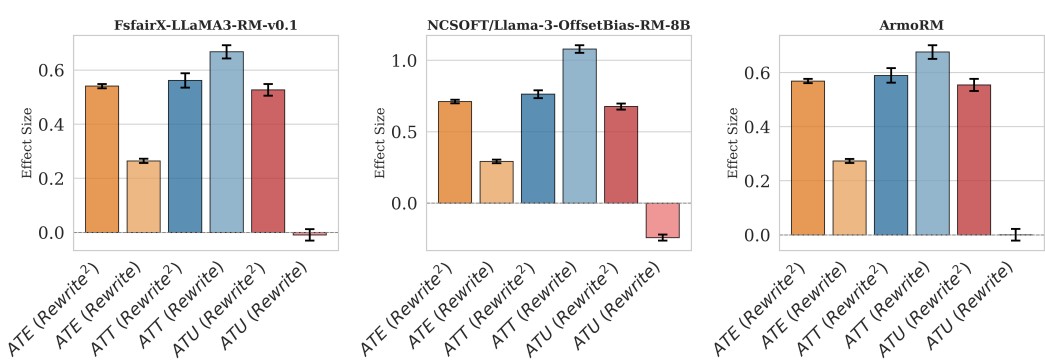

**Figure 10:** Using RATE (rewrites of rewrites) rather than just rewrites changes the estimated treatment effects.

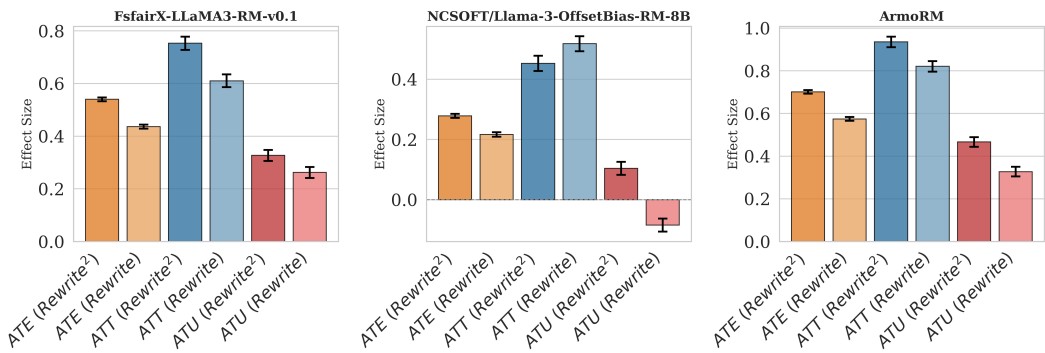

**Figure 11:** Using RATE (rewrites of rewrites) rather than just rewrites changes the estimated treatment effects.

