# OpenReview forum: "RATE: Score Reward Models with Imperfect Rewrites of Rewrites"
_ICLR.cc/2025/Conference — Submitted to ICLR 2025_

### Official Review · Reviewer_PZwU · 2024-10-31

**Soundness:** 2
**Presentation:** 2
**Contribution:** 2
**Rating:** 6
**Confidence:** 5

**Summary:**

The paper proposes a more causal perspective to evaluate reward model by generating pairs of responses are the explicitly rewritten to differ in a particular aspect. To this end they propose a rewrite of rewrite strategy that seems to be most effective in evaluating reward model biases.
They apply their methods on several datasets including semi synthetic ones to illustrate the novelty of their evaluation method.

**Strengths:**

- The paper proposes a new way to construct evaluation data pairs for reward modelling through rewritten which to the best of my knowledge has not been done before.
- The paper also investigates the effects of rewriting the rewrites which I find quite intriguing as well as interesting. They how that rewriting the rewrites do affect the reward model in some way.
- The paper also shows on semi-synthetic data how they method is more robust to distribution shifts than naive methods.

**Weaknesses:**

- The paper is written in a very convoluted way for a relatively simple method.
- Especially the experiments are described in a way that I find extremely hard to follow even after repeatedly reading the section. Hence i have some questions regarding this paper.
1) What are you expecting to see in Figure 2? "The naive estimator overstates the length bias" How do you know the ground truth length bias and how can you claim yours does better? This part is very unclear to me and seems to be the crux of the misunderstanding. If you could clarify I would be more than happy to raise my score.
2) Figures 4 and 5 seem to show the effects of rewriting rewrites. In Figure 4, you see minor changes. In Figure 5 the trend does not seem to be consistent across models. Hence my question is, what are you expecting to see in this figure? what is the ground truth? and how would you pick in practice whether to rewrite the rewrite.
3) Theorem 1 in the paper seems like a standard causal inference setting. and you assume noise assumptions that are just not controllable in the LLM setting. Hence my question to you is what the point of that theorem is? Can you please justify the additive reward structure and what the motivation for that is?

**Questions:**

I have written all my questions above.
Happy to raise my score if the above is addressed well.

---

> ### Author Response · Authors · 2024-11-22
> **Additional Experiment verifying double-rewrite better than single-rewrite**
>
> Thank you for your review, questions, and insights!
>
> These are good questions. To clarify, the overall goals of our experiments are both to demonstrate that RATE is better in semi-synthetic  scenarios where we know the expected causal effect, and to demonstrate that the distinction between RATE and other methods is meaningful in real-world situations (where the true effect is unknown). Individually, the experiments are aimed at:
> 1. Show the RATE estimator (greatly) outperforms the naive estimator in a setting where the correct behavior is known (Fig 3)
> 2. Demonstrate that RATE and the naive (non-counterfactual) estimator are substantively different in the wild (Fig 2)
> 3. Show that the rewriter changes off-target attributes in a manner that the reward model is sensitive to (Fig 4)
> 4. Show that the single-rewrite  method is substantially  different from the double-rewriting method (Fig 5).
>
> These experiments do not directly address whether RATE is \emph{better} than the single rewrite method. We take the main evidence for this to simply be the examples of the rewrites and rewrites-of-rewrites (tables 4, 8-12). It is a clear (albeit subjective, human) judgement that the rewrites and rewrites-of-rewrites are more closely matched than the rewrites and original responses.
>
> To give a more direct test of this, we have now run an additional experiment demonstrating that spurious correlations in off-target features show 1. a treatment effect with a single rewrite that 2. vanishes with a double rewrite. In detail: we measure the effect of a movie review starting with a consonant on rewards. We expect this effect to be 0 (quality is not affected by the review starting with the consonant!). We show that RATE robustly estimates this effect as 0, but that the single rewrite (and naive) methods are highly sensitive to whether there were typos in the original reviews.
>
> Results:
> - NCSOFT: https://tinyurl.com/rateviz1
> - FsfairX: https://tinyurl.com/rateviz2
> - ArmoRM: https://tinyurl.com/rateviz3
>
>
>
> Regarding the weaknesses:
>
> | The paper is written in a very convoluted way for a relatively simple method.
> - The idea of using rewrites of rewrites is itself a simple idea. However, it is not obvious that \emph{when a rewriter is known to produce bad rewrites, that rewriting twice will somehow improve the estimation of the ATE}. The bulk of the paper is dedicated to \emph{verifying} the validity of this simple method and formalizing the intuitions. Section 4 investigates what assumptions about the rewriter (e.g. GPT-4o) needed in order for RATE to accurately estimate the ATE. Meanwhile, the experiments in Section 5 constitute a suite of validation tests. We will emphasize this focus on \emph{validity}, and why such validity is not immediately obvious, in the paper.
>
> | What are you expecting to see in Figure 2...
> - Figure 2 is meant to show the \emph{difference} in effects using a naive estimator and RATE. In this setting, we do not have a ground truth: the point is simply to emphasize that the \emph{choice of estimation procedure} strongly affects the conclusions drawn about e.g. length bias (or any other attribute of reward models). To support our claim that RATE is more robust to spurious correlations in the data than either the naive estimator or a procedure using a single rewrite, please refer to the additional experiment above (as well as Figure 3).
>
> | Figures 4 and 5 seem to show the effects of rewriting rewrites...
> - We have added a new experiment (see above) that directly compares the effectiveness of the double-rewrite method to the single-rewrite method. The advantages of this new experiment over Figures 4 and 5 are threefold: 1. There is a clear ground truth (the true ATE is likely close to 0), 2. The double rewrite method is much more effective at achieving this ground truth compared to the single rewrite method, and 3. This trend is consistent across all three reward models. Based on the results of this paper, we recommend \emph{always} rewriting the rewrite.
>
> | Theorem 1 in the paper seems like a standard causal inference setting...
> - It seems intuitive that the rewrite operation would just work (as well as the double-rewrite). But many things seem intuitive and aren’t true, so we would like some formalization which shows that the intuitive thing actually corresponds to the causal estimand. The value of the theorem is not in being predictive (that’s the role of empirics), but rather in “have you defined things clearly and precisely enough to be able to confirm the causal effect”, which is nontrivial to do. It’s too naive to model the rewrite as being perfectly counterfactual, but under suitable assumptions you don’t need it to be.
> - The additivity assumption isn’t particularly hard to satisfy: for instance, the fact that GPT-4o ~always cleans up typos but doesn’t add new typos satisfies this assumption. We will add clarification in the paper about how common rewriter behaviors fit within these formal assumptions.

---

> > ### Comment · Reviewer_PZwU · 2024-12-02
> > **Thanks for the additional experiments**
> >
> > I want to thank the authors for the additional experiments and clarifications after which the paper makes more sense to me.
> > Hence I will increase my score accordingly.

---

> > > ### Author Response · Authors · 2024-12-02
> > >
> > > Thanks, your review raised some good questions and we think the paper is stronger with the additional experiment.

---

### Official Review · Reviewer_6RUH · 2024-11-04

**Soundness:** 2
**Presentation:** 3
**Contribution:** 3
**Rating:** 6
**Confidence:** 4

**Summary:**

This paper focus on the evaluation of reward models used in language modeling. The authors propose to  use rewrites of rewrites to correct for the bias introduced in intervention of LLM's rewrite.  They evaluate the proposed method and show its effectiveness at correcting for spurious correlations in the data.

**Strengths:**

I appreciate the idea of using causality to evaluate the reward model, especially rewrite twice to address the introduced noise in intervention.

The proposed method are simple, straightforward but effective and may be further helpful in reward hacking.

**Weaknesses:**

See questions.

**Questions:**

The paper is written in a clear and accessible manner, making it easy to understand and follow. Therefore I have only a few minor questions and suggestions.

In Lines 211 and 212, if I understand correctly, $\text{Re}(y^{ij}, 0)$ refers to rewriting $y^{ij}$ such that the corresponding attribute is zero. Should this instead be $\text{Re}(y^{ij}, 1)$, and similarly, should $\text{Re}(y^{ij}, 1)$ be $\text{Re}(y^{ij}, 0)$? Please double-check the notation and explain their reasoning if it is correct as written.

One more thing I am interested in: could you please give more examples of the spurious correlations in the data, except for the length? Please give a brief discussion of how the proposed method could be applied to different types of spurious correlations beyond length.

A couple of suggestions:
- The subscript $i$ in $x^{i}$ could potentially be omitted for brevity if applicable.
- In Figure 1, it might be helpful to represent that the "helpful/unhelpful" state is the true cause of the response, while length serves as a spurious cause within the data. This may be helpful for the readers to understand that we want the reward model learn the actual cause, "helpful/unhelpful", but usually the reward model may learn the spurious cause, "length". See some related paper also investigating the real cause of rewards in LLM [1] and traditional RL [2].

[1] Tien, J.Y., He, J.Z., Erickson, Z.M., Dragan, A.D., & Brown, D.S. (2022). Causal Confusion and Reward Misidentification in Preference-Based Reward Learning. International Conference on Learning Representations.

[2] Zhang, Y., Du, Y., Huang, B., Wang, Z., Wang, J., Fang, M., & Pechenizkiy, M. (2023). Interpretable Reward Redistribution in Reinforcement Learning: A Causal Approach. Neural Information Processing Systems.

---

> ### Author Response · Authors · 2024-11-22
>
> Thank you for your positive feedback!
>
> Questions:
>
> You are correct about the typo in lines 211-212! Thank you for pointing this out, this has been fixed and will be included in the final draft.
>
> | Could you please give more examples of the spurious correlations in the data, except for the length? Please give a brief discussion of how the proposed method could be applied to different types of spurious correlations beyond length.
> - Spurious correlations can arise at any level of abstraction over text data, and are the norm, not the exception; this has motivated the rich field of causal inference over text. Spurious correlations can manifest in various ways, such as sentiment being linked to the genre of movie reviews (e.g., sci-fi reviews often skew positive), the frequency of typos correlating with the document's topic (more typos in restaurant reviews than legal documents).
> - In our experiments, we consider 4 attributes: length, sentiment, complexity, and helpfulness; see Figure 2. Applying RATE to a new concept is straightforward: consider the generic rewriting instruction in Table 2: “Adjust this response so it’s {W}, but change *nothing* else.” Resolve ambiguities in the definition of the concept W, by qualitatively checking rewrites of random samples (see Table 1, Table 2, and Appendix B). Then apply Algorithm 1 to estimate the average treatment effect of W on the reward.
>
> The suggested papers address a separate but related question: why learned reward models may misidentify the true reward function underlying preference data, as it is difficult to determine which factors are responsible for human preferences. Such misidentification would typically be evaluated using the “naive estimate”, which itself suffers from spurious correlations in the evaluation. On the other hand, our method allows for more reliable detection of such misidentification, which could better inform attempts at mitigation. We will clarify this in the related work.

---

> > ### Comment · Reviewer_6RUH · 2024-11-22
> >
> > Thanks for the effort you put into addressing my concerns and clarifying the points raised!

---

### Official Review · Reviewer_yTTN · 2024-11-04

**Soundness:** 2
**Presentation:** 2
**Contribution:** 1
**Rating:** 3
**Confidence:** 4

**Summary:**

This paper proposes to reduce Length bias in preference datasets by having Large Language Models (LLMs) regenerate responses. The authors theoretically prove that under reasonable assumptions, this rewriting can maintain the consistency of preferences. They also demonstrate through experiments that this method can indeed avoid the sensitivity of RMs to irrelevant metrics on both synthetic and real data.

**Strengths:**

- The theoretical formulas are quite detailed.
- The authors' experiments cover three different types of tasks.
- Under the setup of this paper, the ablation studies conducted by the authors are logically consistent.

**Weaknesses:**

- The biggest issue with this paper lies in the validity and applicability of the method.
  - Validity of the method. The authors have made efforts in the experiment section to demonstrate that RATE can reduce the impact of irrelevant factors on RM, and I do not doubt these results. However, these alone are insufficient to support the contribution of the paper. The authors should also show that RATE can simultaneously improve the accuracy of RM scoring. For instance, they should demonstrate the enhancement that RM brings to methods such as BoN, RLHF, RFT, etc., through experiments across multiple tasks and models. Otherwise, it would be difficult for readers to confidently deploy this method in practical RMs based solely on the experimental results presented in the paper, which would significantly undermine the contribution of the paper.
  - Applicability of the method. The authors mention that RATE requires calling gpt-4 to rewrite responses in the preference dataset, indicating that the RATE method relies on expert models for modification. This will greatly limit the scope of application of the method.
- The presentation of experimental results is very confusing.
  - In Figure 2, the authors aim to show that RATE can reduce the length bias of the reward model towards responses. However, from the figure, I observe that the sensitivity of the reward model to factors such as Sentiment, Complexity, and Helpfulness has also decreased. I believe this will raise concerns among researchers about the effectiveness of the RATE method, as it could significantly alter the original performance of the RM.
  - In Figure 3, the authors intend to illustrate that RATE can enhance the invariance of the reward model to irrelevant metrics. However, the persuasiveness of this single figure is quite weak. What people care about is whether the RM can score according to human true preferences. For example, a reward model that always gives a score of 0 to any response can still achieve the effect shown in Figure 3, but this is not what humans actually want.

**Questions:**

- Does RATE affect the accuracy of RM scoring? Is it a positive or negative impact? Are there any quantifiable metrics?
- Can RATE help RM better assist LLMs in alignment? Has this been verified on mainstream methods such as BoN, RLHF, etc.?

---

> ### Author Response · Authors · 2024-11-22
>
> We would like to clarify what the scope of our paper is:
> 1) Our paper is not about reducing length bias in preference datasets. To be clear, RATE does not actually modify reward models. It is a method for estimating properties of the reward model, namely their sensitivity to attributes, formalized as a treatment effect. Nowhere do we state that our goal is to “maintain the consistency of preferences” or “avoid the sensitivity of RMs to irrelevant metric.”
> 2) “ablation studies” is not part of our paper.
>
> Regarding the weaknesses:
> 1) Our method does not attempt to reduce the impact of irrelevant factors on reward models, but rather to address bias in the naive estimate (i.e. without counterfactuals) due to spurious correlations with off-target features.
> 2) This interpretation of figures is not quite right. For Figure 2, we are comparing the RATE estimates vs. the naive estimates across various attributes, not actually modifying the reward model. The reviewer’s comment on Figure 3 also seems to misunderstand our method’s purpose.
>
> Regarding the Questions:
> - We understand these questions to be asking about whether RATE improves the alignment between reward models and preference datasets. As discussed above, RATE does not modify reward models, it is about \emph{accurately measuring} a property of the reward model. It is interesting to consider whether such a measurement tool could be used to improve alignment, but this is secondary to the question of whether that measurement tool can even be trusted, which is the investigation in our paper.

---

> > ### Comment · Reviewer_yTTN · 2024-11-25
> >
> > First of all, I apologize for misunderstanding the author's article. After rereading it carefully, I realized that the contribution of the paper is to use a rewrite-based method to evaluate the scoring accuracy of the RM (Reward Model) on certain metrics.
> >
> > As I understand it, the author seems to have reconstructed an evaluation set based on certain specific dimensions. Compared to traditional evaluation sets, there is a more pronounced difference between \( r_w \) and \( r_l \) on that specific dimension (e.g., helpfulness), while there is no difference on other dimensions. This allows for an independent evaluation of the RM's scoring accuracy on this dimension.
> >
> > If the author believes my understanding is correct, I would like to express my following concerns:
> >
> > 1. It seems that a crucial baseline is missing, which is to directly let an expert model, such as GPT-4o mentioned in the paper, generate responses. For example, adding to the prompt: "Please generate two responses of the same length, one helpful and one not helpful."
> >
> > 2. Should different dimensions be evaluated independently? I believe that slight length bias is beneficial for RM because longer responses indeed provide more detailed information at times. Broadly speaking, is it appropriate to decouple the evaluation of various dimensions of human preference?
> >
> > 3. The author's evaluation method still relies on the expert model GPT-4o. I am curious whether the rewrite method can be applied to models with the same capabilities as the RM. For instance, using a 7B instruction-tuned model to evaluate a 7B RM.

---

> > > ### Author Response · Authors · 2024-11-27
> > >
> > > Thank you for engaging.
> > >
> > > It seems that a misunderstanding persists. We are not evaluating accuracy of reward models. We are measuring how sensitive reward models are to certain attributes---often, this is about undesirable behavior. For example, the fact that alignment often changes the length of responses is generally undesirable, and the method in this paper allows us to assess whether this behavior is due to undesirable behavior of the reward model.
> > >
> > > With respect to your questions:
> > > 1. The goal here is to generate a pair of responses that differ only in the target attribute without needing to enumerate all off-target attributes. This is why re-writing is critical.
> > > 2. We are not clear what you are asking here---can you elaborate?
> > > 3. We note that the only thing matters for the efficacy of the procedure is the quality of the rewrites. So, generally, it seems best to use whatever model produces the most convincing rewrites (as judged by simply reading the output examples). There is no coupling of the language model used for rewriting and the reward model that is evaluated. We did attempt rewriting with 7B-tier models and found they struggled to rewrite responses effectively.

---

### Official Review · Reviewer_F8Gx · 2024-11-09

**Soundness:** 3
**Presentation:** 3
**Contribution:** 3
**Rating:** 5
**Confidence:** 2

**Summary:**

This paper addresses the evaluation of reward models in language modeling by introducing RATE (Rewrite-based Attribute Treatment Estimators), a method for measuring the causal effect of a specific attribute (e.g., length) on the reward assigned to a response. RATE uses large language models to rewrite responses, generating imperfect counterfactuals, and adjusts for rewrite errors by performing a second rewrite. The paper demonstrates the effectiveness of RATE on both synthetic and real-world data.

**Strengths:**

The topic of reward model evaluation is highly relevant and important for large language models.

The paper introduces a new approach, creating response pairs where only the attribute of interest varies through rewrites, enabling causal estimation.

The paper  leverages LLM-based rewrites and rewrites of rewrites to control for biases, an innovative strategy that enhances reliability in causal estimation.

**Weaknesses:**

The paper focuses primarily on response length as an attribute. Are other attributes considered?

The approach's effectiveness may be sensitive to the LLM’s ability to generate accurate counterfactuals, potentially impacting the reliability of causal estimates.

The rewrite instructions appear straightforward; adding prompts specific to target attributes could improve accuracy.

The experiments lack analysis on the effectiveness of the rewrites themselves.

The method depends heavily on the LLM for rewriting; experiments using different LLMs for rewriting could help assess robustness.

**Questions:**

For the rewrite instructions, does the method use the same target model, or is another model employed for implementing the rewrite instructions?

How does the paper assess the effectiveness of the rewrite instructions?

What reward models were used in the evaluation experiments? How sensitive is RATE to the quality and specificity of the initial LLM-based rewrites?

---

> ### Author Response · Authors · 2024-11-22
>
> Thank you for your review!
>
> | The paper focuses primarily on response length as an attribute. Are other attributes considered?
> - In the experiments, we consider 4 attributes: length, sentiment, complexity, and helpfulness; see Figure 2. We will further emphasize this point.
>
> | The approach's effectiveness may be sensitive to the LLM’s ability to generate accurate counterfactuals, potentially impacting the reliability of causal estimates.
> - We agree that the efficacy of the approach relies on the LLM being able to generate counterfactuals. Notice that there are two parts to this. The rewrite must be able to change the targeted concept and must be able to \emph{not} change off-target concepts. The paper is mainly focused on the question of not changing off-target concepts. This is the motivation for the rewrite-of-rewrite structure. With respect to the ability of the model to modify the on-target concept, this can be checked by directly examining the modified outputs!
> - However, we agree that the method is only applicable where rewrites are actually possible---e.g., we cannot test quantities like ‘persuasiveness’ that LLMs can’t effectively generate. We will highlight this as a limitation of the work.
>
> | The method depends heavily on the LLM for rewriting; experiments using different LLMs for rewriting could help assess robustness.
> - It is true that the method is ultimately contingent on the LLM efficacy. However, whether the editing is effective can be checked by simply examining the outputs. Presumably, the procedure will become more powerful as LLMs continue to improve. It is an interesting idea to try to examine the extent to which different LLMs have different propensity for off-target edits, but this is somewhat orthogonal to the aim of the paper.
>
> | For the rewrite instructions, does the method use the same target model, or is another model employed for implementing the rewrite instructions?
> - Another model (GPT-4o) is used for the rewrites, which is different from the reward models being studied.
>
> | How does the paper assess the effectiveness of the rewrite instructions?
> - We randomly sample rewrite samples for qualitative verification, and have two quantitative assessments: 1. Figure 3 where we show with semi-synthetic data that the rewrites are not changing off-target attributes, even though the rewrite instructions do not explicitly list which off-target attributes to avoid changing (as demonstrated by the flat line for RATE). 2. Furthermore, per our reply to reviewer PZwU, we ran a more direct experiment that demonstrates that single rewrites are not as effective at leaving off-target concepts unchanged. We will include this in the paper.
>
> | What reward models were used in the evaluation experiments? How sensitive is RATE to the quality and specificity of the initial LLM-based rewrites?
> - As discussed in Section 5 and depicted in Figure 2, we consider three popular publicly available reward models: FsfairX-LLaMA3-RM-v0.1, NCSOFT/Llama-3-OffsetBias-RM-8B. These were selected because they are top-performing open-source models on RewardBench.

---

> ### Comment · Reviewer_F8Gx · 2024-11-27
>
> It would be better to show the effectiveness of the approach.
>
> How about using different LLM models?

---

> > ### Author Response · Authors · 2024-12-02
> >
> > Could you elaborate on what you mean by "effectiveness"? We interpret this as "whether the rewrite has successfully changed the target concept without changing any off-target concepts", and we have a suite of experiments to demonstrate this effectiveness, as discussed in our previous reply.

---

> > > ### Comment · Reviewer_F8Gx · 2024-12-02
> > >
> > > To be clear, it would be better to use different LLMs for rewriting and compare the results.

---

> > > > ### Author Response · Authors · 2024-12-02
> > > >
> > > > Can you clarify what question such an approach might answer? It seems to us that the only relevant quantity is the quality of the rewrites produced by the language model, which should be judged by simply examining the rewrites. We see no reason to use a weaker LLM for the rewriting (the rewrite cost is quite cheap using the OpenAI API). We wonder if the request for this experiment is actually revealing some other more basic misunderstanding of the aim of the work?
> > > >
> > > > As part of development, we tested rewrites using 8 billion parameter scale LLMs. We found that the rewrites produced by these models were of relatively poor quality (because of weak language understanding). We will add example samples to the output and a reproduction of the synthetic example tasks.

---

### Meta-Review · Area_Chair_6cw9 · 2024-12-21

**Metareview:**

This paper addresses the evaluation of reward models in language modeling, focusing on the challenge of these models being imperfect proxies for actual preferences. It introduces RATE, a method that measures the causal effect of response attributes on assigned rewards by using large language models to create counterfactuals and adjust for rewriting errors.

Despite the work's contributions, this paper still has to be improved on the sensitivity of its approach to LLM accuracy, the clarity of rewrite instructions, the analysis of rewrite effectiveness, robustness across different LLMs, the validity of its method, the confusing presentation of experimental results, and the justification for its assumptions and theorem.

**Additional Comments On Reviewer Discussion:**

The author and reviewers discussions are multi-turns and some of them are thorough.

---

### Decision · Program_Chairs · 2025-01-22

Reject